# One-dimensional Magnus-type platinum double salts

Christopher H. Hendon[1,2], Aron Walsh[1,3], Norinobu Akiyama[4,†], Yosuke Konno[4,†], Takashi Kajiwara[5], Tasuku Ito[6], Hiroshi Kitagawa[7] & Ken Sakai[8,9,10]

Interest in platinum-chain complexes arose from their unusual oxidation states and physical properties. Despite their compositional diversity, isolation of crystalline chains has remained challenging. Here we report a simple crystallization technique that yields a series of dimer-based 1D platinum chains. The colour of the $Pt^{2+}$ compounds can be switched between yellow, orange and blue. Spontaneous oxidation in air is used to form black $Pt^{2.33+}$ needles. The loss of one electron per double salt results in a metallic $d_{z^2}$ state, as supported by quantum chemical calculations, and displays conductivity of $11\,S\,cm^{-1}$ at room temperature. This behaviour may open up a new avenue for controllable platinum chemistry.

[1] Department of Chemistry, Centre for Sustainable Chemical Technologies, University of Bath, Claverton Down, Bath BA2 7AY, UK. [2] Department of Chemistry, Massachusetts Institute of Technology, Cambridge, Massachusetts 02139, USA. [3] Department of Materials Science and Engineering, Yonsei University, Seoul 03722, South Korea. [4] Faculty of Science, Department of Applied Chemistry, Science University of Tokyo, Kagurazaka, Shinjuku-ku, Tokyo 162-8601, Japan. [5] Faculty of Science, Department of Chemistry, Nara Women's University, Kitauoyanishi-machi, Nara 630-8506, Japan. [6] Department of Chemistry, Graduate School of Science, Tohoku University, Sendai 980-8578, Japan. [7] Division of Chemistry, Graduate School of Science, Kyoto University, Kitashirakawa-Oiwakecho, Sakyo-ku, Kyoto 606-8502, Japan. [8] Faculty of Science, Department of Chemistry, Kyushu University, Motooka 744, Nishi-ku, Fukuoka 819-0395, Japan. [9] International Institute for Carbon-Neutral Energy Research (WPI-I2CNER), Kyushu University, Motooka 744, Nishi-ku, Fukuoka 819-0395, Japan. [10] Center for Molecular Systems (CMS), Kyushu University, Motooka 744, Nishi-ku, Fukuoka 819-0395, Japan. † Present addresses: Fujisoft Incorporated, Kandaneribeicho 3, Chiyoda-ku, Tokyo 101-0022, Japan (N.A.); Nippon Kayaku CO., Ltd, Sanyo Onoda, Yamaguchi 757-8686, Japan (Y.K.). Correspondence and requests for materials should be addressed to A.W. (email: a.walsh@bath.ac.uk) or to K.S. (email: ksakai@chem.kyushu-univ.jp).

In 1828, Magnus[1] reported a one-dimensional (1D) Pt-chain compound that featured alternating $[PtCl_4]^{2-}$ and $[Pt(NH_3)_4]^{2+}$ building blocks ($d_{Pt-Pt} = 3.25$ Å), Magnus' green salt. However, it was not until the turn of the twentieth century that in-depth studies into the structure and reactivity of similar organometallic Pt chains began[2]. In 1908, the first $Pt^{3+}$-containing compound was discovered through the treatment of cis-PtCl₂(acetonitrile)₂ with a silver salt (for example, Ag₂SO₄)[3]. The material was described as 'platinblau' (platinum blue) because of its characteristic dark blue colour and indirect evidence suggested a polymeric/oligomeric structure involving Pt–Pt interactions[4,5]. Yet, despite the intriguing colour and nearly a century of enthusiastic efforts by chemists following the discovery, both the crystal and electronic structures of platinblau remain an enigma.

In the late 1960s, a series of conductive (0.1 S cm$^{-1}$) 1D Pt chains composed of either $[Pt(CN)_4]^{2-}$ or $[Pt(ox)_2]^{2-}$ (ox = oxalate; Fig. 1) were developed by Krogmann[6]. The properties of these materials were eventually realized to be a product of partial removal of electrons from the 1D Pt chain[7]. For the tetranuclear Pt-chain compounds (considered as partial structures of Krogmann's salts), the one-electron oxidized species (that is, $Pt_4^{2.25+}$) resulted in a blue compound reminiscent of the platinblau; other Pt-chain compounds with different nuclearity and oxidation states were also observed[8–12]. Then in 2006, Brédas and colleagues[13] revisited Magnus' green salt ([Pt(NH₃)₄][PtCl₄]) and showed that the material possesses interesting band properties for conductive applications, owing to the Pt–Pt interactions; later, Drew et al.[14] harnessed similarly highly coloured Pt-chain complexes as selective photochromic sensors.

In a $Pt^{2+}$-chain complex, the highest occupied state is constructed by an alternating array of antibonding Pt $d_z^2$ orbitals. Oxidation can result in either delocalization of the missing electron over many sites ($Pt^{[2+\delta]+}$)[15,16] or formal oxidation from $Pt^{2+} \rightarrow Pt^{3+}$ (ref. 17). In the past half century, a variety of highly coloured platinum complexes[18] have been isolated including red[19], orange[20], yellow[21,22], green[23,24] and blue[25] Pt-chain materials. In the case of tetranuclear platinum complexes, these colours arise from a variety of oxidation states including $Pt^{2+}$ (Pt(II)₄), $Pt^{2.25+}$ (Pt(II)₃Pt(III)), $Pt^{2.5+}$ (Pt(II)₂Pt(III)₂) and $Pt^{3+}$ (Pt(III)₄, although the fully oxidized systems tend to behave as $Pt_2^{3+}$ dimers with two axial donors).

Around the same time as Krogmann's developments, $Pt^{2+}$ complexes found application in medicinal chemistry, propelled by

the anticancer treatment cis-platin[26–28]. These works provided an interesting avenue for platinum chemistry and subsequent focus shifted away from the unusual electronic properties associated with Pt-chain complexes towards molecular reactivity, with complexes often featuring relatively simple ligands similar to those shown in Fig. 2. However, fundamental research into ligand design continued alongside this more applied chemistry.

The amidate ligand (Fig. 1) is interesting, because it produces doubly bridged dimeric Pt₂ building blocks. A typical formula is $[Pt(II)_2(NH_3)_4(\mu\text{-amidate})_2]^{2+}$ (amidate can also be α-pyridonate, α-pyrrolidonate, 1-methyluracilate, 1-methylthyminate, acetaminidate and so on)[8–12]. Tetranuclear complexes arise from a stack of two dimeric units where the dimer–dimer interaction is stabilized by ligand-mediated quadruple hydrogen bonds formed between the O(amidate) and N–H(amine) units. In such cases, the coordination manner of two amidates must be in a head-to-head (HH) arrangement, permitting a stack of N₂O₂-ligated Pt coordination planes. Moreover, the Pt–Pt bonding interactions within the tetranuclear units are reinforced by shortening the Pt–Pt distances as the oxidation state increases. Both the tetrenuclear and octanuclear species exhibit the characteristic partial oxidation associated with the colourful Pt-chain complexes[29,30].

With the recent progress in conductive metal-organic framework chemistry[31–36], interest in highly conductive hybrid materials has been reignited. Metal-organic materials typically do not have high intrinsic electrical conductivity, because the metals are spatially separated by organic ligands[37]. These 1D Pt chains pose interesting pathways to access highly ordered and conductive metal-organic wires[38]. There has only been one dimer-based conductive 1D Pt family[39]. The material was electrochemically grown in oxidizing conditions and produced a black Pt-chain complex ($[Pt_2(NH_3)_4(\mu\text{-acetate})_2]^{2+}$) that, similar to Krogmann's Pt salts, was susceptible to partial oxidation at the metal centres. This material demonstrated a moderate electrical conductivity of 0.001–0.01 S cm$^{-1}$, despite its relatively poor crystallinity.

Here we report the formation of a series of $Pt^{2+}$ Magnus-type double salts: 1D $Pt^{2+}$ chain composed of alternating cationic (+/+) and anionic (2-) $Pt^{2+}$ building blocks. Our initial target was a stoichiometric mixture of $[HT\text{-}Pt(II)_2(bpy)_2(\mu\text{-pivalamidate})_2]^{2+}$ (1, HT indicates a head-to-tail arrangement of the pivalamidate ligands) and $[Pt(II)(ox)_2]^{2-}$ (2); however, the result is a new family of compounds. We detail the crystallization procedure, followed by characterization of their crystal structure and physical properties including colour and electrical conductivity.

## Results

**Crystal growth and characterization.** We developed a simple kinetically controlled crystallization procedure to provide access to hitherto unknown Pt-chain systems. The petri-paper three-zone crystallization method (Fig. 2) isolates the reagents, located in zones 1 and 3, from the crystallization region, zone 2. In practice, this is achieved by using filter paper separating the zones, thereby slowing the rate of salt diffusion. Aqueous 1 and 2 were added to zones 1 and 3, respectively. After 2 days, bright yellow crystals of $[HT\text{-}Pt(II)_2(bpy)_2(\mu\text{-pivalamidate})_2][Pt(II)(ox)_2]$ (3), were isolated and were found to crystallize in a polymeric $Pt^{2+}$-chain complex (Fig. 3a).

As observed for the monomeric 1 (that is, the diplatinum cation $[HT\text{-}Pt(II)_2(bpy)_2(\mu\text{-pivalamidate})_2]^{2+}$)[40] (Supplementary Fig. 1), the yellow crystalline material 3 undergoes the HT→HH isomerization over 24 h through the gradual dissolution of 3 to form the compositionally identical 4

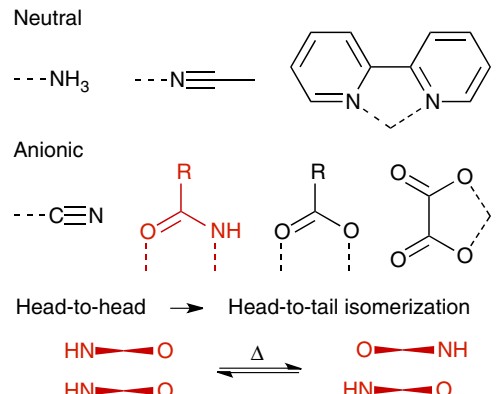

**Figure 1 | Ligands commonly employed in Pt-chain chemistry:** Examples of familiar neutral and anionic ligands found in Pt-chain complexes. The bridging amidate ligand, shown in red, can form hydrogen bonds in the direction of the Pt chain, stabilizing the Pt centres and promoting oxidation.

Neutral

Anionic

Head-to-head → Head-to-tail isomerization

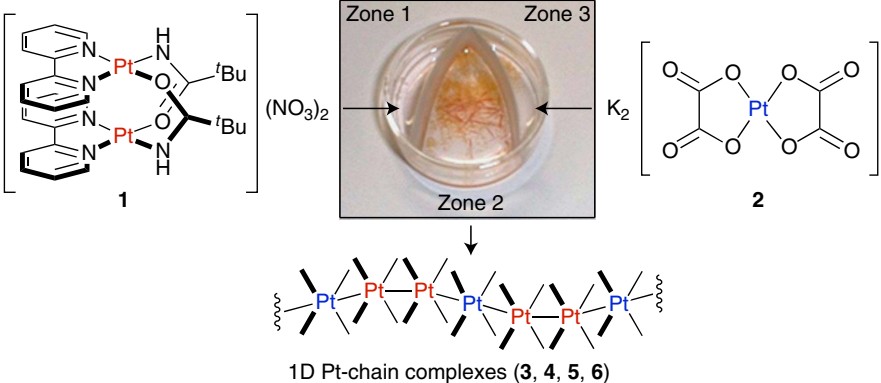

**Figure 2 | The petri-paper three-zone crystallization procedure.** Aqueous solutions of **1** and **2** are added to zones 1 and 3, respectively. Slow diffusion through the filter paper into the crystallization zone 2 yields large and highly crystalline **3**, with **4**, **5** and **6** resolved over time.

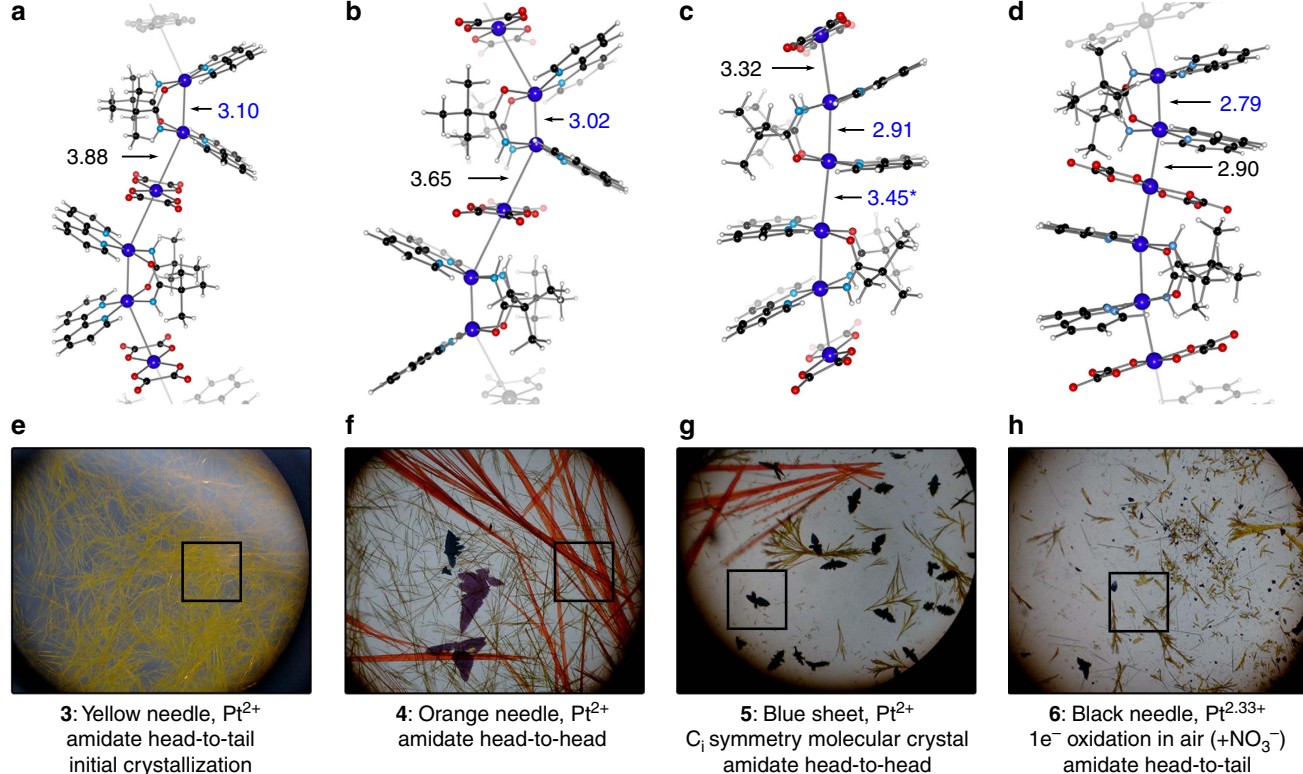

**3:** Yellow needle, $Pt^{2+}$ amidate head-to-tail initial crystallization

**4:** Orange needle, $Pt^{2+}$ amidate head-to-head

**5:** Blue sheet, $Pt^{2+}$ $C_i$ symmetry molecular crystal amidate head-to-head

**6:** Black needle, $Pt^{2.33+}$ $1e^-$ oxidation in air ($+NO_3^-$) amidate head-to-tail

**Figure 3 | Products isolated from the crystallization method:** Sequential crystallization of Pt double salts. Structure **3**, shown in **a**, is a yellow needle (**e**) polymeric material. Over 24 h, **3** undergoes a HT→HH isomerization forming **4**, shown in **b**, orange needles. (**f**) Simultaneously, a discrete hexamer (**c**) with HH isomerism forms, **5** shown in **g**. Another purple sheet-like material is formed, shown by optical microscopy in **f**; it appears to be analogous to **5** with a different stacking manner or with different isomerism of the dimer unit, but the crystallinity proved insufficient for characterization. After exposure to air over several weeks, partially oxidized polymeric black needles of **6** with HT isomerism are formed (**d,h**). No HH mixed-valence compounds have been isolated. The Pt–Pt separations are in units of Å. In the case of **3**, **4** and **5**, $H_2O$ is omitted for clarity. Both $H_2O$ and $NO_3^-$ are omitted from the presentation of **6**.

([HH-Pt(II)$_2$(bpy)$_2$($\mu$-pivalamidate)$_2$][Pt(II)(ox)$_2$]·5.5H$_2$O) (bright orange needles shown in Fig. 3b). Relative to **3**, the pivalamidate-bridged Pt–Pt distances decrease by ∼0.1 Å, which is consistent with the preference of HT-Pt(II)$_2$ dimers for a bridged Pt–Pt distance longer than the corresponding HH-Pt(II)$_2$ dimers, probably due to higher electrostatic repulsion given within a symmetric dimeric structure having two N$_3$O-ligated Pt geometries[24].

Remembering that the highest occupied states of Pt$^{2+}$-chain complexes are $\sigma^\star(d_{z^2} - d_{z^2})$[41], the colours associated with Pt

compounds are principally correlated with the lowest-energy optical transition, corresponding to the so-called metal-metal-to-ligand charge transfer transition. Compound **4** has shorter Pt–Pt bonds, leading to greater destabilization of the $\sigma^\star$, and hence to the red shift in transition energy. In addition, the adjacent bipyridine ligands interact, lowering the lowest unoccupied molecular orbital energy proportional to inter-Pt distance (resulting in a red shift with HT→HH isomerization).

During the pivalamidate HT→HH isomerization and subsequent crystallization of **4**, a structurally dissimilar blue

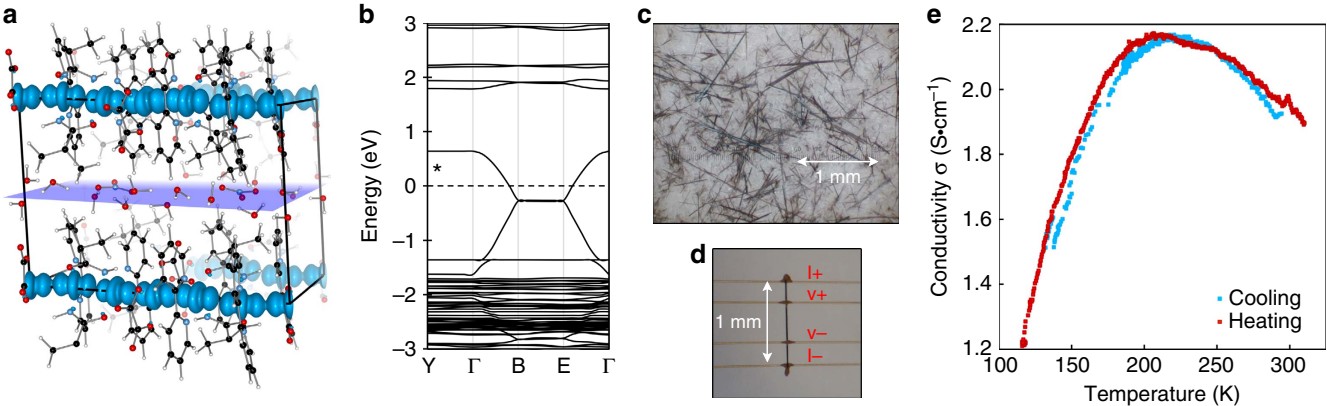

**Figure 4 | Electronic properties of compound 6.** [HT-Pt$_2$(bpy)$_2$($\mu$-pivalamide)$_2$][Pt(ox)$_2$](NO$_3$) · 7H$_2$O, **6**, where the calculated spin density (blue isosurface at 0.02 e/Å$^3$) is delocalized along the Pt chain in the <001> direction. The hydration layer solvates (NO$_3^-$)$_2$ in [200], shown as purple plane (**a**). The electronic band structure (**b**) depicts the metallic character in the direction of the delocalized Pt$^{2.33+}$ chain. The addition of 2e$^-$ (denoted by *) increases the Fermi level (dotted line) to fill the valence band, resulting in a semiconducting band gap analogous to **3**. The target synthesis of **6** resulted in large black needles (**c**), which could be mounted directly for conductivity measurements (**d**). Using gold paste contacts provided a peak conductivity of 11 S cm$^{-1}$, whereas the carbon paste used in the temperature sweeping measurement (**e**) produced a peak conductivity of 2.2 S cm$^{-1}$ at ~210 K. O, H, N, C and Pt are depicted in red, white, blue, black and grey, respectively.

material simultaneously formed (shown in the optical microscope image; Fig. 3c and Supplementary Fig. 13). This structure was determined to be discrete hexamers of [HH-Pt(II)$_2$(bpy)$_2$ ($\mu$-pivalamide)$_2$]$_2$[Pt(II)(ox)$_2$]$_2$ · 8H$_2$O (**5**). From the immense amount of work on Pt-blue complexes, we assumed that this was a mixed-valence material that had partially oxidized in the presence of air[42]. However, the formulation of **5**, clearly solved by crystallography as a pair of **1** and **2**, allows us to conclude that **5** is a hexameric Pt$^{2+}$ species with no paramagnetic character (confirmed by magnetic susceptibility measurements and a lack of electron spin resonance (ESR) response). Compound **5** is a rare example of a Pt$^{2+}$-blue compound and can be envisaged as the dimerization of two trimeric species. The resultant hexamer possesses a crystallographic inversion centre at which a stack of N$_2$O$_2$-coordinated Pt planes is realized in the same manner as established in various tetranuclear Pt-blue-related species.

The terminal Pt–Pt bonding observed in **5** provides one potential explanation for the blue chromophore (making **5** appear similar to the Pt$_4^{2.25+}$ blue species). The terminal Pt–Pt associations in the hexamer exhibit exceptionally shorter inter-Pt distances than any other inter-monomer–dimer associations. This indicates that dative bonding from the filled $\sigma$* towards part of the vacant molecular orbitals (such as the Pt 6p$_z$ in [Pt(ox)$_2$]$^{2-}$) is dominant, leading to manifestation of net partial oxidation of the inner Pt$_4$ geometry.

Crystallization of Magnus-type double salts continued to evolve over 14 days: **3**, **4** and **5** dissolved and recrystallized into the terminal material, fine black needles, **6** (shown in Fig. 3d). Single crystal X-ray diffraction (XRD) confirmed the formation of [HT-Pt$_2$(bpy)$_2$($\mu$-pivalamide)$_2$][Pt(ox)$_2$](NO$_3$) · 7H$_2$O. This stoichiometry represents a 1e$^-$ partial oxidation per double salt in the presence of air (that is, delocalized Pt$^{2.33+}$). Compound **6** is structurally analogous to **3**, with the addition of charge balancing NO$_3^-$ solvated in the [200] disordered aqueous layer, as shown by the purple plane in Fig. 4a. Dramatic shortening occurs in both the bridged and non-bridged Pt–Pt distances, reflecting partial oxidation at all Pt centres including those ligated with oxalates. Only the HT was observed, in part due to crystallographic requirement: a crystallographic twofold axis is passing through the midpoint of the Pt–Pt bond within the dimeric unit, that is, this dimeric unit is considered to possess a

crystallographic HT isomerism. The HH analogue would result in anisotropic localization of the electrons (such as the blue species, **5** with discrete Pt centres). Following the classification of Robin and Day[43], these are class IIIB mixed-valence compounds with highly delocalized electrons.

## Discussion

Unlike the colourful Pt-chain complexes, black compounds are interesting for electronic applications, owing to their potentially high conductivity and strong optical absorption. Before this, there has only been one Pt-black chain complex reported[39]. Quantum chemical calculations were used to elucidate the electronic structure of **6**. The unpaired valence electrons are delocalized along the Pt chain (Fig. 4a) in agreement with its diamagnetic character evidenced by electron spin resonance spectroscopy. Furthermore, the homogeneous charge distribution about the Pt nuclei is in agreement with X-ray photoemission spectroscopy measurements (Supplementary Figs 10 and 11d, and Supplementary Table 5). From the electronic band structure (Fig. 4b), we deduce that **6** is metallic with a partially occupied band along the Pt$^{2.33+}$ chain. In the reduced state (that is, electronically similar to compound **3**), the addition of two electrons per crystallographic unit cell (that is, one electron per Magnus stack) would increase the Fermi level (dotted line) to fill the valence band, forming a material with a finite band gap.

For electrical transport characterization, single crystal conductivity measurements were performed. The crystallinity and yield of **6**, as formed from the sequential crystallization, did not result in large enough crystals to perform such measurements. A revised target synthesis was designed, promoting partial oxidation of **2** to motivate the direct formation and isolation of **6** (yield = ca. 30%, >1 mm black needles; Fig. 4c). The conductivity was measured by mounting a single crystal of **6** (Fig. 4d) on four electrodes. The crystal was carefully attached using gold paste and a champion conductivity of 11 S cm$^{-1}$ was obtained at room temperature. To investigate the temperature dependence of the electrical transport gold paste could not be used, because it resulted in cracking of crystal with sweeping temperature. Carbon paste results in a decrease in maximum conductivity to 2.2 S cm$^{-1}$, but the measurements identified a metal-semiconductor transition at ~210 K (Fig. 4e).

Our study has provided several advances in the synthetic chemistry and property control of Pt salts. First, we demonstrated that controlled crystallization using the petri-paper procedure provides a cheap and accessible method for the recovery of novel structures. Identification of a blue $Pt^{2+}$ compound suggests that the elusive platinblau may not require non-integral oxidation states. We further showed that oxidation of this double salt to $Pt^{2.33+}$ results in a black highly conductive material, as supported by *ab initio* calculations that revealed a metallic state with delocalized electrons at the Fermi level. This provides an important design principle for developing conductive metal-organic networks, where redox reactions can be used to install conductivity postsynthetically. Our results highlight several important questions including whether we can harness the $Pt^{2+} \rightarrow Pt^{3+}$ oxidation for catalysis and whether these double salts can provide an alternative to molecular analogues for $Pt^{2+}$-based medicines? We anticipate that these findings will stimulate further interest in fundamental platinum chemistry.

## Methods

**Synthetic approach.** [HT-Pt(II)$_2$(bpy)$_2$($\mu$-pivalamidate)$_2$](NO$_3$)$_2 \cdot$ 5H$_2$O, **1**, was prepared as previously described by Yokokawa and Sakai[40], and K$_2$[Pt(ox)$_2$] $\cdot$ 2H$_2$O, **2**, was prepared as previously described by Werner and Grebe[2]. Using the petri-paper three-zone method, aqueous solutions of **1** and **2** were added to zones 1 and 3, respectively. The crystals were then developed unsealed (that is, in air), at room temperature, within a few days, sequentially crystallizing as [HT-Pt(II)$_2$(bpy)$_2$($\mu$-pivalamidate)$_2$][Pt(II)(ox)$_2$] (**3**, yellow needles), [HH-Pt(II)$_2$(bpy)$_2$($\mu$-pivalamidate)$_2$][Pt(II)(ox)$_2$] $\cdot$ 5.5H$_2$O (**4**, orange needles) and [HH-Pt(II)$_2$(bpy)$_2$($\mu$-pivalamidate)$_2$]$_2$ [Pt(II)(ox)$_2$]$_2 \cdot$ 8H$_2$O (**5**, dark blue plates, hexaplatinum species). Representative crystals were removed from the reaction mixture and physical property measurements were performed on them. Leaving of the mixture for 1–2 weeks led to the formation of [HT-Pt$_2$(bpy)$_2$ ($\mu$-pivalamidate)$_2$][Pt(ox)$_2$](NO$_3$) $\cdot$ 7H$_2$O (**6**, fine black needles), a partially oxidized species. Although this sequential procedure resulted in very low yeids, a refined target synthesis was devised. The improved synthetic strategy involved the following: (i) oxidation by persulfate, (ii) ageing of crystals at higher temperature and (iii) manual separation of well-formed needles from the relatively small crystals deposited simultaneously. In this method, a solution of **2** (0.04 mmol) and K$_2$S$_2$O$_8$ (0.02 mmol) in water (12 ml) was heated at 65 °C for 1 h, followed by leaving it at room temperature for 1 h. To the solution, **1** (0.04 mmol, as solid) was added and the temperature was gradually raised to 65 °C over 1 h without stirring. The mixture was further left at 65 °C for 6 h to grow well-formed black metallic needles of **6**, crystallographically identical to the fine needles obtained in the other procedure. Finally, the solution was very slowly cooled down to room temperature over 6 h. As the crystals obtained possessed relatively wide distribution in size, only the needles longer than 180 μm were collected through tedious manual sieving. The resulting crystals were then washed with a minimum amount of water and dried in air (yield: ca. 30 %). Full characterization details, including single-crystal XRD, $^{195}$Pt NMR, X-ray photoemission spectroscopy, thermogravimetric analysis (TGA), solid-state ultraviolet–visible and conductivity measurements are described in the Supplementary Figs 1–16, Supplemental Tables 1–5, Supplemental Notes 1–4 and Supplementary Data 1–4.

**Computational approach.** All calculations were performed within the Kohn–Sham density functional theory framework using periodic boundary conditions, to approximate the infinite salts, starting from the crystallographic unit cells determined from the XRD measurements. The Vienna *ab initio* simulation package[44], a planewave code (with PAW scalar relativistic core potentials)[45], was employed for all geometry optimizations and electronic calculations. For each system, the lattice vectors and internal atomic positions were relaxed with the semi-local Perdew–Burke–Ernzerhof exchange-correlation functional revised for solids[46]. For these calculations, a 500-eV plane-wave cutoff basis set was found to be suitable for convergence of the systems to within 0.01 eV per atom. To provide quantitative electronic information, non-local hybrid density functional theory calculations were performed using the HSE06 functional[47], with 25% of the short-range semi-local exchange functional replaced by the exact non-local Hartree–Fock exchange. Visualizations of the structures and orbitals were made using the codes VESTA[48].

**Data availability.** The input structures for the solid-state quantum mechanical calculations are available from https://github.com/WMD-group/Crystal_structures. The X-ray crystallographic coordinates for structures reported in this study have been deposited at the Cambridge Crystallographic Data Centre (CCDC), under deposition numbers CCDC 1477434-1477437. These data can be obtained free of charge from the Cambridge Crystallographic Data Centre via www.ccdc.cam.ac.uk/data_request/cif.

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

## Acknowledgements

This work was supported by a Grant-in-Aid for Scientific Research on Priority Areas ('Metal-assembled Complexes') from the Ministry of Education, Culture, Sports, Science and Technology of Japan. This was partly supported by the International Institute for Carbon Neutral Energy Research (WPI-I2CNER) sponsored by the World Premier International Research Center Initiative (WPI), MEXT, Japan. Computations benefited from access to the High Performance Computing Consortium, which is funded by EPSRC Grant EP/L000202. Additional support has been received from the Royal Society and the ERC (Grant 277757) and the NSF-funded XSEDE facilities (Grant ACI-1053575).

## Author contributions

All compounds were prepared by Y.K. and K.S. The petri-paper method was invented by Y.K. and K.S.. Improved synthesis of compound **6** was developed by N.A. T.I., T.K. and K.S. performed the crystallographic measurements and structure refinement. Electrical conductivity measurements were carried out by N.A., K.S. and H.K.. C.H.H. and A.W. performed the density functional theory calculations. C.H.H., A.W. and K.S. wrote the paper.

## Additional information

**Competing financial interests:** The authors declare no competing financial interests.

