## [Peer review file · Nature Communications]

Reviewers' Comments:

Reviewer #1 (Remarks to the Author)

Sakai and co-workers report an interesting work on One-dimensional Magnus Type Platinum double salts. The experimental work has been carried out with due diligence and the rationalization of the obtained results has been discussed well. The theoretical calculations are top-notch. The manuscript has been written well for most part. The introduction is found to be more wanting and could be drastically improved. Although I find the results in the manuscript interesting, the novelty is quite lacking and how far the approach is a generalized one is also a bigger question. The authors have omitted a great chunk of previously reported work by the groups of Smith (example: Adv.Mat. 2006, pp 2039) and Mann (example: Chemistry of Materials (2009), pp 3117-3124). Based on this above mentioned publications and a plethora of other ones, the novelty is certainly diminished to a great extent. The lack of novelty of the results as well as the lack of the results catering to a broader audience, the manuscript at the current stage is not sufficient enough to warrant publication in Nature communications.

Reviewer #2 (Remarks to the Author)

Structures

The authors have reported four interesting, and challenging structures. The clear, detailed explanation of the refinement of structure 6 in the "X-ray Crystallography Measurements and Software" section of the Supplementary Information document and Figure S6 is commended. A similar treatment of the other three structures in the SI document would lead to a significant improvement, and address many of the comments in this review. In order to address the comments below further refinements of structure 4 may be necessary.

The CheckCIF report does not include the full set of tests as the system was not able to extract structure factors from the CIF (even though they are present). To remedy this without using a more recent version of ShelXL it will be necessary to upload the .fcf files directly to CheckCIF (and submit them with your manuscript).

The CheckCIF reports for 3, 4 and 5 all contain the alert "Implicit Hall Symbol Inconsistent with Explicit C 21 c1". This should be corrected.

Structure 3

It would be helpful if some comment/explanation could be given in the "X-ray Crystallography Measurements and Software" section of the Supplementary Information document concerning the indicators of slightly poorer dataset quality for this structure, i.e. the somewhat high Rint for this dataset, the low C-C bond precision and slightly high residual electron density.

The N-H hydrogen atom has been fixed, and does not form any hydrogen bonds. If possible it would be better if this hydrogen atom could have been refined, as proof of its' existence in the structure. If this was not possible (understandable given the somewhat high Rint value) then this should be discussed in the "X-ray Crystallography Measurements and Software" section of the Supplementary Information document.

Structure 4

It seems that O13 and O19 are one of the oxalate oxygen atoms disordered over two sites, at 50 % occupancy per site. However, the C36-O19 bond is far too long and the O19 atom is out of the

plane. The treatment of this disorder thus needs to be addressed, which will probably require further refinements.

The CheckCIF alert "_symmetry_space_group_number does not match H-M symbol" can be corrected, but it is slightly concerning that this error should have occurred in the first place.

The N-H hydrogen atoms have been fixed. They do not form any hydrogen bonds and form somewhat close intramolecular distances with other hydrogen atoms. Given these concerns then, if possible it would be better if these hydrogen atoms could have been refined, as proof of their existence in the structure. If this was not possible (which would be understandable given the complexity of this structure) then this should be discussed in the "X-ray Crystallography Measurements and Software" section of the Supplementary Information document and the justification for their inclusion in the structure explained. (I am sure there is a perfectly valid justification for their inclusion, but it would just be good for this to be given in the SI document).

Structures 4 and 5

Most of the serious CheckCIF alerts relate to the fact that no hydrogen atoms have been refined for the solvent water molecules. While this is understandable, it is important that an explanation is provided preferably both in the CIF and the "X-ray Crystallography Measurements and Software" section of the Supplementary Information document.

An explanation of the disorder in these structures should be provided in the "X-ray Crystallography Measurements and Software" section of the Supplementary Information document.

Manuscript

Page 2, line 138: "Figure 2c" should read "Figure 3c" and "Figure S8" is probably referring to Figure S4.

Page 2, line 165: "Figure 2d" should read "Figure 3d".

Caption, Figure 3: There is no mention that solvent and counter-ion molecules have been omitted.

Page 3, lines 175-176: It is not clear what is meant by "(in part due to crystallographic requirement)".

Caption, Figure 4: I do not see the asterix (*) referred to in panel b.

Page 4, lines 243-252 and _publ_section_experimental section of the CIF: While this section of the manuscripts states that 3, 4 and 5 crystallised sequentially at room temperature in air within a few days comments in the CIF suggest that crystals of 4 and 5 were unstable in air, losing water solvate to such a degree that they had to be encapsulated in a capillary for data collection. It seems likely that it was the removal of the crystals from their aqueous mother liquor that led to the de-solvation and so it would be clearer if line 243 could mention the existence of the mother liquor. At the moment line 243 states that the crystals grew "in air" which, when considering the presence of the mother liquor is not strictly the case.

Supplementary Information

Page 2, "X-ray Crystallography Measurements and Software" section: It is stated that all datasets were collected at 296 K but, from the CIF that for 4 was collected at 200 K.

Page 2, "X-ray Crystallography Measurements and Software" section: From the CIF it seems that the absorption correction for the dataset from 6 was different to that carried out for the datasets from 3, 4 and 5. This should be made clearer.

Page 2, "X-ray Crystallography Measurements and Software" section: SX-X is used. Presumably this should refer to actual figures.

Page 2, "X-ray Crystallography Measurements and Software" section: This section should include the information given in the `_publ_section_experimental` section of the CIF; that the crystals of 4 and 5 had to be encapsulated in a capillary during data collection due to solvent loss.

Page 3: The empirical formula of 6 is given as $[\text{Pt}(2.33+)_2(\text{bpy})_2(\mu\text{-pivalamidate})_2][\text{Pt}(2.33+)(\text{ox})_2](\text{NO}_3)_{1.07} \cdot 7.34\text{H}_2\text{O}$ while the calculation for the nitrogen content per three platinum atoms result in the figure of 1.089 (compared to 1.07 given in the empirical formula). Is this an error or is there a reason for the difference?

Captions for Figures S3, S4 and S5: These should contain the same level of detail as that for Figure S6, including the probability level of the ellipsoids and the identity of anything omitted for clarity (hydrogen atoms, solvent molecules etc).

CIF

The reference for PLATON has not been given in `_publ_section_references`.

Reviewer #3 (Remarks to the Author)

There is some very interesting coordination chemistry in this paper. The crystallization techniques are good and the crystallographic analyses elegant. However the paper is not as easy to read as it could be.

A summary table would help to clarify what complexes have been synthesised and their properties in relation to their structures. The paper would be strengthened by more mechanistic information on the oxidations.

Abstract

"The withdrawal of one electron per double salt"

Not clear what 'withdrawal' means.

What is the mechanism of oxidation?

Line 126

The colours associated with Pt-compounds is

The coloursare

Lines 222-3

"Identification of a blue Pt^{2+} compound suggests that the elusive platinblau could be a simple divalent species"

The older term valency is still used but for metal ions oxidation state or oxidation number is better.

The literature suggests that mixed oxidation states and the presence of Pt(III) is a feature of platinum blues. This statement implies that platinum blues might only contain Pt(II) which certainly makes the paper interesting.

Have any EPR studies been done in the present case to check for the presence of Pt(III) in any of the complexes?

Line 234

"We anticipate that these findings will stimulate renewed interest in platinum chemistry."

I do not understand this sentence. In fact there is currently already enormous interest in platinum chemistry related to its use in anticancer drugs. More than 50% of all cancer chemotherapies involve a platinum compound.

It is a pity there are not some initial results for the potential applications hinted at in the conclusion:

"Our results highlight several important questions including whether we can harness the Pt²⁺/Pt³⁺ oxidation for catalysis, and if these double-salts can provide an alternative to molecular analogues for Pt²⁺-based medicines?"

Are the complexes all soluble enough to be used as medicines? If so more solution studies might be profitable.

Reviewer #1: Sakai and co-workers report an interesting work on One-dimensional Magnus Type Platinum double salts. The experimental work has been carried out with due diligence and the rationalization of the obtained results has been discussed well. The theoretical calculations are top-notch. The manuscript has been written well for most part. The introduction is found to be more wanting and could be drastically improved. Although I find the results in the manuscript interesting, the novelty is quite lacking and how far the approach is a generalized one is also a bigger question. The authors have omitted a great chunk of previously reported work by the groups of Smith (example: Adv.Mat. 2006, pp 2039) and Mann (example: Chemistry of Materials (2009), pp 3117-3124). Based on this above mentioned publications and a plethora of other ones, the novelty is certainly diminished to a great extent. The lack of novelty of the results as well as the lack of the results catering to a broader audience, the manuscript at the current stage is not sufficient enough to warrant publication in Nature communications.

Author Response: We thank the reviewer for their favourable response to our research efforts. This work is the product of several years of careful work. We apologise for missing some sections of the literature – Pt chemistry represents a large field with a long history. The highlighted references are now incorporated into the introduction and discussion. The articles referred to be the reviewer are on the topics of conductivity in Magnus' original green salt, and then an application of a similar Pt-chain which exhibited a chromatic response in the presence of aromatics. We would argue that these do not effect the novelty, but rather highlight the rich diversity of research in the area. While there are now many reports of conductive Pt chains, none show the complex HT-HH isomerization or the variation of colour over a large range in a single family that we have discovered. We have further emphasized the novel aspects in the revised text.

Reviewer #2

Author Note: We preface this response with the apology for the uploading on the incorrect cif files during the review process. We have now provided the correct files with reports, and the comments/concerned posed by this reviewer are remedied.

The authors have reported four interesting, and challenging structures. The clear, detailed explanation of the refinement of structure 6 in the "X-ray Crystallography Measurements and Software" section of the Supplementary Information document and Figure S6 is commended. A similar treatment of the other three structures in the SI document would lead to a significant improvement, and address many of the comments in this review. In order to address the comments below further refinements of structure 4 may be necessary. The CheckCIF report does not include the full set of tests as the system was not able to extract structure factors from the CIF (even though they are present). To remedy this without using a more recent version of ShelXL it will be necessary to upload the .fcf files directly to CheckCIF (and submit them with your manuscript).

Author Response: All *.cifs and their CheckCIF reports have been renewed.

The CheckCIF reports for 3, 4 and 5 all contain the alert "Implicit Hall Symbol Inconsistent with Explicit C 21 c1". This should be corrected.

Author Response: These problems have been solved in the revised materials. We apologize for the mix up with the cif files during submission, and the inconveniences due to this.

Structure 3. It would be helpful if some comment/explanation could be given in the "X-ray Crystallography Measurements and Software" section of the Supplementary Information document concerning the indicators of slightly poorer dataset quality for this structure, i.e. the somewhat high Rint for this dataset, the low C-C bond precision and slightly high residual electron density. The N-H hydrogen atom has been fixed, and does not form any hydrogen bonds. If possible it would be better if this hydrogen atom could have been refined, as proof of its' existence in the structure. If this was not possible (understandable given the somewhat high Rint value) then this should be discussed in the "X-ray Crystallography Measurements and Software" section of the Supplementary Information document.

Author Response: Usually, refinement of H atoms in the presence of heavy platinum atoms is not possible. In our previous work, we have shown that the binding directions of amidates in such doubly bridged platinum dimers can be rationally determined by comparing the results of refinement for two possible directions. It is often possible to determine the binding direction of the O and NH moieties of the amidate group by comparing the results of least-squares calculations performed for two possible directions. We have now given the more detailed explanations on these issues.

Structure 4. It seems that O13 and O19 are one of the oxalate oxygen atoms disordered over two sites, at 50 % occupancy per site. However, the C36-O19 bond is far too long and the O19 atom is out of the plane. The treatment of this disorder thus needs to be addressed, which will probably require further refinements.

Author Response: We now supply the details in ESI, explaining the validity of our models.

The CheckCIF alert "_symmetry_space_group_number does not match H-M symbol" can be corrected, but it is slightly concerning that this error should have occurred in the first place.

Author Response: This error has been solved. We apologize for this inconvenience.

The N-H hydrogen atoms have been fixed. They do not form any hydrogen bonds and form somewhat close intramolecular distances with other hydrogen atoms. Given these concerns then, if possible it would be better if these hydrogen atoms could have been refined, as proof of their existence in the structure. If this was not possible (which would be understandable given the complexity of this structure) then this should be discussed in the "X-ray Crystallography Measurements and Software" section of the Supplementary Information document and the justification for their inclusion in the structure explained. (I am sure there is a perfectly valid justification for their inclusion, but it would just be good for this to be given in the SI document).

Author Response: Thank you for the kind suggestions. After our extensive refinement trials for this structure, we now judge that the HH dimer is disordered in its NO binding direction, meaning that the HH dimeric unit itself is disordered in its direction of N→O. Thus, all the four bridging amidate donor sites are occupied by both N and O atoms with each having an occupation factor of 0.5. At each site, the N and O atoms are refined by applying common atomic coordinates and temperature factors. Only by using this refinement scheme, all the amidate N and O atoms have been found to possess moderately balanced equivalent temperature factors (see ESI for details).

Structures 4 and 5. Most of the serious CheckCIF alerts relate to the fact that no hydrogen atoms have been refined for the solvent water molecules. While this is understandable, it is important that an

explanation is provided preferably both in the CIF and the "X-ray Crystallography Measurements and Software" section of the Supplementary Information document.

Response: Hydrogen atoms of water molecules in the crystals of such platinum complexes are usually not found in the difference Fourier map. Refinement usually fails. Therefore, we usually keep them unlocated. These are now explained in ESI.

An explanation of the disorder in these structures should be provided in the "X-ray Crystallography Measurements and Software" section of the Supplementary Information document.

Author Response: We now provide the details in ESI.

Page 2, line 138: "Figure 2c" should read "Figure 3c" and "Figure S8" is probably referring to Figure S4.

Author Response: We thank the reviewers for this comment, indeed Figure 2c should read Figure 3c, and Figure S8 should actually read Figure S13. This has been corrected in text.

Page 2, line 165: "Figure 2d" should read "Figure 3d".

Author Response: Fixed.

Caption, Figure 3: There is no mention that solvent and counter-ion molecules have been omitted.

Author Response: The caption has been updated to reflect that we only are presenting the Pt-chain part of the material.

Page 3, lines 175-176: It is not clear what is meant by "(in part due to crystallographic requirement)".

Author Response: Since a crystallographic two-fold axis is passing through the midpoint of the Pt-Pt bond within the dimeric unit, this dimeric unit is considered to possess a crystallographic HT isomerism. We have carefully thought about this presentation of this problem and revised the text accordingly.

Caption, Figure 4: I do not see the asterix (*) referred to in panel b.

Author Response: We apologise for this. It was lost underneath a layer in the image. The * should now be visible in the updated manuscript.

Page 4, lines 243-252 and `_publ_section_experimental` section of the CIF: While this section of the manuscripts states that 3, 4 and 5 crystallised sequentially at room temperature in air within a few days comments in the CIF suggest that crystals of 4 and 5 were unstable in air, losing water solvate to such a degree that they had to be encapsulated in a capillary for data collection. It seems likely that it was the removal of the crystals from their aqueous mother liquor that led to the de-solvation and so it would be clearer if line 243 could mention the existence of the mother liquor. At the moment line 243 states that the crystals grew "in air" which, when considering the presence of the mother liquor is not strictly the case.

Author Response: This is a very insightful comment and indeed the reviewer is correct. We have amended this section accordingly to reflect that the crystals were isolated throughout the growing process.

Supplementary Information

Page 2, "X-ray Crystallography Measurements and Software" section: It is stated that all datasets were collected at 296 K but, from the CIF that for 4 was collected at 200 K.

Author Response: It is now stated in ESI that the data collection of 4 was done at 200 K. We apologize for this error.

Page 2, "X-ray Crystallography Measurements and Software" section: From the CIF it seems that the absorption correction for the dataset from 6 was different to that carried out for the datasets from 3, 4 and 5. This should be made clearer.

Author Response: We apologize for this error. We have ascertained that the reviewer is correct. Reflection data for 3-5 were done using SADABS, which is now clearly stated in the revised ESI.

Page 2, "X-ray Crystallography Measurements and Software" section: SX-X is used. Presumably this should refer to actual figures.

Author Response: This has been corrected to now read Figure S3-6.

Page 2, "X-ray Crystallography Measurements and Software" section: This section should include the information given in the _publ_section_experimental section of the CIF; that the crystals of 4 and 5 had to be encapsulated in a capillary during data collection due to solvent loss.

Author Response: These are now described in the ESI.

Page 3: The empirical formula of 6 is given as [Pt(2.33+)2(bpy)2(μ -pivalamidate)2][Pt(2.33+)(ox)2](NO3)1.07·7.34H2O while the calculation for the nitrogen content per three platinum atoms result in the figure of 1.089 (compared to 1.07 given in the empirical formula). Is this an error or is there a reason for the difference?

Author Response: This part has been revised based on the real refinement strategy. The details of the disorder model adopted at the end are now carefully described, also by adding some pictures describing the model in the ESI.

Captions for Figures S3, S4 and S5: These should contain the same level of detail as that for Figure S6, including the probability level of the ellipsoids and the identity of anything omitted for clarity (hydrogen atoms, solvent molecules etc).

Author Response: Figures S3-S6 are now just showing the ORTEP diagrams together with some selected distances and the symmetry operations. Instead of showing the crystallographic data on these figures, we have added Table S2 showing crystallographic data of all four compounds.

CIF. The reference for PLATON has not been given in _publ_section_references.

Author Response: We have included reference for Spek (PLATON) in all cifs now.

Reviewer #3: There is some very interesting coordination chemistry in this paper. The crystallization techniques are good and the crystallographic analyses elegant. However the paper is not as easy to read as it could be. A summary table would help to clarify what complexes have been synthesised and their properties in relation to their structures. The paper would be strengthened by more mechanistic information on the oxidations.

Abstract

"The withdrawal of one electron per double salt"

Not clear what 'withdrawal' means.

What is the mechanism of oxidation?

Author Response: We have replaced "withdrawal" by "loss". The mechanism is explained in the text.

Line 126. The coloursare

Author Response: Fixed

Lines 222-3

"Identification of a blue Pt²⁺ compound suggests that the elusive platinblau could be a simple divalent species" The older term valency is still used but for metal ions oxidation state or oxidation number is better.

Author Response: The term has been replaced

The literature suggests that mixed oxidation states and the presence of Pt(III) is a feature of platinum blues. This statement implies that platinum blues might only contain Pt(II) which certainly makes the paper interesting. Have any EPR studies been done in the present case to check for the presence of Pt(III) in any of the complexes?

Author Response: Preliminary EPR was indeed performed on these materials and the hexameric Pt-Blue compound was found to be EPR silent.

Line 234 "We anticipate that these findings will stimulate renewed interest in platinum chemistry." I do not understand this sentence. In fact there is currently already enormous interest in platinum chemistry related to its use in anticancer drugs. More than 50% of all cancer chemotherapies involve a platinum compound.

Author Response: We agree and have rephrased it to "stimulate further interest in fundamental platinum chemistry"

It is a pity there are not some initial results for the potential applications hinted at in the conclusion: "Our results highlight several important questions including whether we can harness the Pt²⁺ Pt³⁺ oxidation for catalysis, and if these double-salts can provide an alternative to molecular analogues for Pt²⁺-based medicines?" Are the complexes all soluble enough to be used as medicines? If so more solution studies might be profitable.

Author Response: This is an important point and indeed our next aim. The solution chemistry is complex. We touched upon this issue in 1998 - "New Structural Aspects of Pyrrolidinonate-Bridged and Pyridonate-Bridged, Homo- and Mixed-Valence, Di- and Tetranuclear cis-Diammineplatinum Complexes: Eight New Crystal Structures, Stoichiometric 1:1 Mixture of Pt(2.25⁺)₄ and Pt(2.5⁺)₄, New Quasi-One-Dimensional Halide-Bridged [Pt(2.5⁺)₄-Cl•••]_n System, and Consideration for Solution Properties", J. Am. Chem. Soc., 1998, 120, 8366-8379. Solubility can be controlled but the effects on the redox chemistry can be challenging to predict. This research will keep us busy in the near term.

Reviewers' Comments:

Reviewer #1 (Remarks to the Author)

The authors have submitted a revised version of the previously submitted manuscript. Several changes have been made that has enhanced the manuscript. Appropriate citations have been included with the novelty clearly stated better in this version of the manuscript. Given the overall changes made, I recommend acceptance of the manuscript to Nature communications as it stands.

Reviewer #2 (Remarks to the Author)

The Authors are congratulated on the improvements made to the structures and structural descriptions in this submission and their thorough response to the comments made. These were clearly very challenging structures which have been analysed with a great deal of care. There is just one small correction needed in the Supplementary Information document. In the first paragraph of the "X-ray Crystallography Measurements and Software" section reference is made to Figures S3-S6, whereas due to the addition of some extra figures this should probably now read "Figures S6-S9".

Reviewer #3 (Remarks to the Author)

This paper is now suitable for publication. The authors have dealt with my main concerns.